# Uncertain-CAM: Uncertainty-Based Ensemble Machine Voting for Improved COVID-19 CXR Classification and Explainability

**DOI:** 10.3390/diagnostics13030441

**Published:** 2023-01-26

**Authors:** Waleed Aldhahi, Sanghoon Sull

**Affiliations:** School of Electrical Engineering, Korea University, 145 Anam-ro, Seongbuk-gu, Seoul 02841, Republic of Korea

**Keywords:** deep learning, explainable AI, COVID-19, intelligent signal processing, uncertainty

## Abstract

The ongoing coronavirus disease 2019 (COVID-19) pandemic has had a significant impact on patients and healthcare systems across the world. Distinguishing non-COVID-19 patients from COVID-19 patients at the lowest possible cost and in the earliest stages of the disease is a major issue. Additionally, the implementation of explainable deep learning decisions is another issue, especially in critical fields such as medicine. The study presents a method to train deep learning models and apply an uncertainty-based ensemble voting policy to achieve 99% accuracy in classifying COVID-19 chest X-rays from normal and pneumonia-related infections. We further present a training scheme that integrates the cyclic cosine annealing approach with cross-validation and uncertainty quantification that is measured using prediction interval coverage probability (PICP) as final ensemble voting weights. We also propose the Uncertain-CAM technique, which improves deep learning explainability and provides a more reliable COVID-19 classification system. We introduce a new image processing technique to measure the explainability based on ground-truth, and we compared it with the widely adopted Grad-CAM method.

## 1. Introduction

The COVID-19 pandemic, caused by the Severe Acute Respiratory Syndrome Coronavirus 2 (SARS-CoV-2), is considered the worst pandemic in over a century. It has resulted in unprecedented social and economic disruption globally. It was deemed a pandemic in March 2020 by the World Health Organization. Since then, researchers and physicians have put in several efforts to facilitate the early identification of coronavirus symptoms. According to PubMed [1], 755 research articles with the keyword “coronavirus” were published in 2019, and this number further increased to 1245 in the first 80 days of 2020. Due to an incomplete understanding of the viral genome at the beginning of the pandemic, early identification procedures for the disease were crude. The Real-Time Reverse Transcription Polymerase Chain Reaction (RT-PCR) test, also known as COVID-19 RT-PCR, is considered an optimum standard for detecting COVID-19. The RT-PCR is a qualitative method for detecting SARS-CoV-2 nucleic acids present in respiratory specimens at the time of infection [2]. However, there are still numerous difficulties in COVID-19 detection, including the low sensitivity rate of RT-PCR, which is only 60–70%, along with the high costs associated with the test [3]. As COVID-19 primarily manifests as a lung infection [4], the computed tomography (CT) and chest X-rays (CXRs) have been widely employed for its identification. Artificial intelligence and deep learning techniques have been frequently used to detect COVID-19 infection from CT and CXR images. The deep learning techniques are highly popular because they have outperformed conventional methods. The performance of deep learning algorithms has been demonstrated in various scientific applications, such as computer vision [5], object tracking [6], face identification [7], and surgical skill assessment [8]. In contrast to conventional and machine learning techniques, the training features need to not be selected manually. A model can be trained to learn the best features from the dataset used by adjusting the parameters and convolutional neural network (CNN) architecture. Even before the coronavirus pandemic, deep learning techniques were widely integrated in medical imaging. However, since the pandemic, the popularity of the deep learning techniques has increasingly dramatically, and there has been a subsequent rise in the applications in diverse fields, including medical image processing, data science approaches for pandemic modeling, AI for text mining and natural language processing, AI and the Internet of Things, AI in computational biology and medicine [9], and for COVID-19 detection [10]. Because the main characteristics of COVID-19 lung infection are the consolidations and areas of ground-glass opacity (GGO) [11], CXR datasets are suitable to test algorithms for spotting COVID-19 and other lung diseases. The CNNs are at the forefront of illness classification for diagnosis, as well as region of interest segmentation in medical images owing to their ability to accurately learn local features from a specific medical image, such as a CT scan or chest radiograph. A common strategy used to improve the classification performance is to combine the results of various classifiers to produce the final output. The output scores of the individual classifiers are used to combine ensemble techniques. These classifiers may possess multiple architectures for capturing various data items or input vectors produced from the same data instance. Existing methodologies primarily focus on achieving high-performance accuracy without adequately elaborating on what CNNs actually learn or delving into fundamental CNN problems, such as overconfidence predictions and uncertainty evaluations, which are crucial factors in the field of medical diagnosis.

In this study, we propose a novel weighted average ensemble method based on uncertainty to combine the confidence scores of various CNN models and subsequently maximize their efficiency for both COVID-19 classification and explanation. The miscalibration of predictive models is an underrated issue in CNN, and we have capitalized on this situation and converted this issue into an advantage through uncertainty quantification that is measured using PICP as ensemble voting weight. Effective ensembles require a diverse set of skillful ensemble members with varying distributions of prediction errors. In our training framework, we suggested the integration of the cyclic cosine annealing approach with cross-validation to perform voting for each network for accurate predictions of each network before final voting. The output of our training framework exhibits relatively low generalization error and improved performance compared to other frameworks.

The remainder of this paper is organized as follows: Section 2 explains the proposed approach, Section 3 presents some of the experimental results, Section 4 discusses the results, and Section 5 concludes the paper. The main contributions of this study are:A novel CNN training scheme to maximize ensembled model performance and minimize generalization error.A novel uncertainty evaluation ensemble method.A novel uncertainty-based Gradient-weighted Class Activation Mapping (Grad-CAM) ensemble explanation to for a better explanation of the CNN decisions compared to Grad-CAM.A new image processing technique to study the feasibility of the proposed Uncertain-CAM and compare it to normal Gras-CAM.

The proposed approach is evaluated using the COVID-QU-Ex dataset [12].

## 2. Related Literature

Driven by the significant advancements in the CNN technology, research in computational medical imaging has focused on exploring the potential of CNN in medical images produced by CT, magnetic resonance imaging (MRI), and X-rays. Numerous researchers have investigated deep learning technology as a viable method for classification, detection, and segmentation in various medical domains, such as the diagnosis of glaucoma [13], COVD-19, and brain tumors [10,14,15]. In recent years, the application areas of CNN-based AI systems have broadened, and this may further accelerate the analysis of diverse medical images. This is to address the urgent need for differentiating COVID-19 from other pneumonia infections more quickly and accurately. Although new CNN architectures can be built, a high number of images under each class would be required to enhance efficiency.

A recent study [10] found that most research articles employ transfer learning techniques, and only a few rely on fine-tuning, and barely any studies suggest a unique CNN architecture with performance that is on par with that of transfer-learning-based techniques. Transfer learning from models that have been pretrained on the ImageNet dataset has been used in a majority of the studies [16]. The DarkcovNet model, wherein DarkNet19 was used as a classifier for the YOLO object detection system, was proposed for automatic COVID-19 detection using CXR imaging [17]. They reported classification accuracies of 98.08% for binary classification and 87.02% for multiclass classification (COVID-19, pneumonia, and normal) (COVID-19 vs. normal). Polat et al. adopted a debiasing-data-loader strategy and improved the performance of the transfer learning model to avoid the bias problem and acquire high accuracy values for the automatic detection of COVID-19 cases from CXR images [18]. To classify the recovered features into normal, pneumonia, and COVID-19 classes, local binary pattern-based and image-based features were extracted from CXR pictures [19]. According to the authors, the classification of CXR images had a sensitivity of 97.86%. Based on the segmentation of the lung shape before training, an effective deep learning model was proposed to identify COVID-19, which achieved an accuracy of 96.43% [20].

Chowdhury et al. proposed an ensemble approach was proposed for COVID-19 detection in 2021 [21]. Their method takes several snapshots of the same model, and based on the softmax scores of each snapshot, it applies both the hard (majority voting) and soft (score averaging) ensembles for the final prediction. Another ensemble approach for COVID-19 detection, which used three standard CNN models for training and then averaged their softmax scores for the final prediction, was proposed [22].

An uncertainty-aware framework, which utilized transfer learning to detect COVID-19 using X-ray and CT images, was presented in 2021 by Shamsi et al. (2021) [23]. Different pretrained CNNs were used in their proposed framework to extract deep features from CXR and CT images. Additionally, they used different machine learning and statistical modeling techniques to diagnose COVID-19 and gauge the epistemic uncertainty of the categorization outcomes. The MixMatch semi-supervised framework, introduced by Calderon et al., was aimed at enhancing the model uncertainty estimation for COVID-19 identification [24]. Numerous methods have been proposed for estimating the uncertainty in a test, including MC dropout, deterministic uncertainty quantification, and softmax scores. A deep learning system, RCoNetK, was also suggested by Dong et al. for COVID-19 detection and uncertainty estimation [25]. They used multi-expert uncertainty-aware learning, mutual information maximization, and deformable mixed high-order moment features. The RCoNetK model successfully classified X-ray images into COVID-19, pneumonia, and normal classes, and it yielded a sensitivity of 97.76%.

A majority of the previous studies primarily focused on achieving high-performance accuracy or quantifying uncertainty, and less attention was given to explain what the machine actually learned. The main issue associated with CNN, which is overconfidence prediction, was not addressed sufficiently. Motivated by these shortcomings, we have developed a reliable ensemble deep learning network for a more efficient COVID-19 classification and explanation.

## 3. Materials and Methods

In this section, we briefly review the proposed method. First, we outline the benchmark dataset and then describe the proposed architecture and training methodology. Further, we discuss the ensemble modeling techniques and our proposed explainability method, Uncertain-CAM. All these techniques were combined to produce the suggested analytic pipeline, shown in Figure 1 (block diagram). Reproducibility is available at our repository https://github.com/waleed-aldhahi/uncertain-cam/ (accessed on 20 November 2022).

### 3.1. Data Preparation

The highly afflicted countries originally made only modest efforts to share clinical and radiographic data publicly because of the pandemic’s developing nature. Thus, two datasets, COVID-QU and QaTa-Cov19 databases, were produced by a team of researchers from the Qatar and Tampere Universities [26,27]. While the QaTa-Cov19 dataset includes 2951 COVID-19 CXR, along with their ground-truth infection masks, the COVID-QU dataset includes 3616 COVID-19, 8851 non-COVID cases, and 6012 normal cases. Moreover, additional X-rays were gradually becoming available in the public domain. This led to the creation of COVID-QU-Ex [12], which is an expanded version of these datasets and contains over 33,000 CXR pictures from three distinct classes:11,956 COVID-19 samples11,263 cases of pneumonia caused by viruses or bacteria that are not COVID-1910,701 normal (healthy) samples

This dataset was produced and dispersed in different formats that were publicly accessible datasets and repositories. Through a strict quality control procedure, duplicates, incredibly poor-quality, and overexposed photos were found and eliminated, ensuring the high quality of the dataset. The generated dataset consists of images with a high level of interclass heterogeneity and few variations in resolution, quality, and SNRA. A detailed explanation of different data sources is given in [12].

### 3.2. Ensemble Learning

Ensemble-based methods are frequently used in various image-classification tasks, as well as several COVID-19 detection approaches. In summary, majority voting, averaging, and weighted averaging of the predictions that are generated by the classifiers considered for constructing the ensemble are the most commonly employed strategies. In most instances, these methods significantly enhance the overall performance. However, it should be noted that these methods do not consider the accuracy of the prediction when producing an outcome [28].

Ensemble learning supports the “wisdom of crowds” notion, which holds that judgments or predictions by large groups of people are often superior to that of a single expert. Similarly, ensemble learning describes a collection (or ensemble) of basic learners or models that collaborate to produce a more accurate final prediction. Due to excessive variation or strong bias, it is likely that a single model, also referred to as a base or weak learner, underperforms when used separately. However, weak learners can be combined to generate a strong learner, and they outperform any individual base model because their combination minimizes bias or variation [29,30].

Let S be a sample space containing samples $x_{i}\left(i=1,2,\ldots ,s\right)$ belonging to classes $c_{j}\left(j=1,2,\ldots ,N\right)$ which compose the class set $C=\left\{c_{1},c_{2},\ldots ,c_{N}\right\}$. Let K independent voters (basic classifiers) be represented as $f_{k}\left(k=1,\;2,\;\ldots ,K\right).$ Given sample $x_{i}$, the prediction (vote) by $f_{k}$ is $f_{k}\left(x_{i}\right)$. Then, the simple ensemble voter (classifier) system is described as
(1)$$F\left(x_{i}\right)=\{\begin{array}{c} c_{j^{*}}\;if\;\exists \;c_{j^{*}}\in C\;\wedge \;T_{F}\left(x_{i}\in \;c_{j^{*}}\right)=\underset{1\leq j\leq N}{\max}T_{F}\left(x_{i}\in \;c_{j^{*}}\right)\;\\ \;\wedge \;T_{F}\left(x_{i}\in \;c_{j^{*}}\right)>\alpha \times K,\\ c_{reject}\;Else\; \end{array}$$
(2)$$T_{F}\left(x_{i}\in \;c_{j}\right)=\overset{K}{\underset{K=1}{\sum }}y_{k}\left(x_{i}\in \;c_{j}\right),\;j=1,2,\ldots ,N,$$
(3)$$y_{k}\left(x_{i}\in \;c_{j}\right)=\{\begin{array}{c} 1\;if\;f_{k}\left(x_{i}\right)=c_{j}\left(j=1,2,\ldots ,N\right),\;\\ 0\;Else\; \end{array}$$
where $c_{j^{*}}$ represents the predicted class, α×K represents the threshold for voting, and ∧ is the logic operator AND. In this approach, each voter will possess the same weight regardless of their performance. However, by examining the strength of each voter in recognizing and assigning different weights, the ensemble system will yield accurate judgments and a much better performance. The ensemble weighted majority voting system is described as follows:(4)$$F\left(x_{i}\right)=c_{j^{*}}\;if\;\exists \;c_{j^{*}}\in C\;\wedge \;T_{F}\left(x_{i}\in \;c_{j^{*}}\right)=\underset{1\leq j\leq N}{\max}T_{F}\left(x_{i}\in \;c_{j^{*}}\right),$$
(5)$$T_{F}\left(x_{i}\in \;c_{j}\right)=\sum_{K=1}^{K}w_{k}\left(x_{i}\in \;c_{j}\right),j=1,2,\ldots ,N,\;$$
(6)$$V_{winner}\left(x_{i}\in c_{j^{*}}\right)=\left\{f_{1,\;}f_{2},\;\ldots ,f_{k}\right\},$$
(7)$$W\left(x_{i}\in c_{j^{*}}\right)=\left\{w_{1},w_{2},\ldots ,w_{k}\right\},$$
where $w_{k}\left(x_{i}\in \;c_{j}\right)$ denotes the weight of the basic voter $f_{k}$ voting to $c_{j}$ for a given $x_{i}$ and $W\left(x_{i}\in c_{j^{*}}\right)$ represents all the weight lists of the majority votes of winner voters $V_{winner}\left(x_{i}\in c_{j^{*}}\right)$. For optimum learning, we suggest the use of varying training data with k-fold cross-validation for each voter, and for each fold model m, we create s snapshots. The aim of creating model snapshots is to train a single model while continuously lowering the learning rate to reach a local minimum and save a snapshot of the weight of the current model. Further, to move away from the current local minimum, it will be essential to actively accelerate the learning process. The process continues until all cycles are completed. The use of cyclic cosine annealing, aimed at producing model snapshots for CNN, helps gather multiple models during a single training session [31]. The cyclic cosine annealing approach begins with the initial learning rate, progressively drops to the bare minimum, and then shoots back up quickly. Each epoch’s learning rate for cyclic cosine annealing is given by the following expression:(8)$$\alpha \left(t\right)=\frac{\alpha_{0}}{2}\left(\cos\pi \left(\frac{mod(t-1,\left[T/s]\right)}{\left[T/s\right]}\right)+1\right),$$
where T is the count of the training iterations, s is the count of cycles, α(t) is the learning rate at epoch t, and $\alpha_{0}$ is the initial learning rate. The weight of the snapshot model is defined as the weight at the bottom of each cycle. The next learning rate cycle uses these weights but permits the learning algorithm to reach various conclusions, producing a variety of snapshot models. Here, s model snapshots were obtained in s training cycles, and each snapshot was used for ensemble prediction. By using s model snapshot predictions, we applied simple average ensemble predictions for each fold model m of each base classifier (voter). The output of the simple average ensemble predictions for each voter was used to obtain the final weighted voting ensemble, as illustrated in Algorithm 1 and Figure 1. In the next section, we discuss the use of different policies as voting weights.

**Algorithm 1.** Machine weighted voting algorithm.Inputs: Entire Data set D={(x,y):x∈ℝn×p,y∈ℝn}; K base learning algorithmsOutput: Machines Voteinitialization;Split D into DTrain, DTestfor k=1,…,K dofor i=1,…,m splits doSplit DTrain into Ditrain, DiValidation for the ith split.Basic classifier fk train on Ditrain and validate on Divalidationfor j=1,…,S snapshots doCreate S snapshots of fkVjk=Predict on DiValidationConcatenate S predictions on DiValidation, Y^jk=(V1k,…,VSk)Compute the simple average ensemble predictions EikEndConcatenate m predictions Y^k=(E1k,…,Emk)EndUse Y^k′s to compute optimal voting weights (w1*,…,wk*)Apply weighted voting ensemble on DTest (4).End

### 3.3. Optimal Voting Weights

The weighted average ensembles give some models in the ensemble more credit during the prediction because they assume that they are more skilled than the rest. The weighted average or weighted sum ensemble is an improvement over voting ensembles wherein it is assumed that all models are equally competent and contribute proportionally to ensemble predictions. Each model is given a specific weight that is multiplied by the prediction it makes, and it is utilized in the calculation of the sum or average prediction. The difficulty associated with the use of such an ensemble is finding model weights that yield a performance that is superior to both an ensemble with equal model weights and any contributing models. In this section we explore different techniques used to find the optimal voting weights beside our proposed novel method.

#### 3.3.1. Best Combination

The simplest and possibly the most exhaustive approach is to initiate grid searching for weight values between zero and one for each ensemble member such that the weights across all ensemble members add to one. However, according to Shahhosseini et al. [29], using the optimization model proposed by Perrone et al. [32], which aims to combine predictions from fundamental classifiers by determining the ideal weight to aggregate them such that the resulting ensemble reduces the overall Expected Prediction Error (*MSE*), is a more effective approach compared to other methods:(9)$$Min\;MSE\left(w_{1}{\hat{y}}_{1}+w_{2}{\hat{y}}_{2}+\ldots +w_{k}{\hat{y}}_{k},\;Y\right),\;s.t.\;\overset{k}{\underset{k=1}{\sum }}w_{k}=1,\;w_{k}\geq 0,\;\forall f_{k}\left(k=1,\;2,\;\ldots \;,K\right)$$
where ${\hat{y}}_{k}$ denotes the vector of the basic classifier $f_{k}$’s out-of-bag predictions on the validation samples of cross-validation, Y is the vector of true response values, and $w_{k}$ is the weights corresponding to the base model $f_{k}\left(k=1,\;2,\;\ldots \;,K\right)$. The optimization model is described in (10), and it is assumed that n is the total number of instances, $y_{i}\;$is the true value of observation i, ${\hat{y}}_{ik}$ is the base model $f_{k}$, and i is the prediction of observation i.
(10)$$Min\;\frac{1}{n}\overset{n}{\underset{n=1}{\sum }}{\left(y_{i}-\overset{k}{\underset{j=1}{\sum }}w_{k}{\hat{y}}_{ik}\right)}^{2},\;s.t.\;\overset{k}{\underset{j=1}{\sum }}w_{k}=1,\;w_{k}\geq 0,\;\forall f_{k}\left(k=1,\;2,\;\ldots \;,K\right)$$

The formulation is a non-linear convex optimization problem. The computation of the Hessian matrix reveals that the objective function is convex because the constraints are linear. Therefore, this solution is globally deemed the best solution because the local optimum of a convex function (the objective function) on a convex feasible region (the feasible region of the preceding formulation) is guaranteed to be a global optimum [33].

#### 3.3.2. Priori Recognition Performance Statistics

A basic classifier is assigned more weight based on how well it recognizes patterns [30]. Let the confusion matrix of voter *f_k_* be
(11)$$CM_{k}=\left[\begin{array}{cccc} n_{11}^{k} & n_{12}^{k} & \cdots & n_{1N}^{k}\\ n_{21}^{k} & \ddots & \cdots & n_{2N}^{k}\\ \vdots & \vdots & n_{j_{1}j_{2}}^{k} & \vdots \\ n_{N1}^{k} & n_{N2}^{k} & \cdots & n_{NN}^{k} \end{array}\right]\;\left(k=1,\;2,\;\ldots \;,K\right)$$
when $j_{1}=j_{2}$, $n_{j_{1}j_{2}}^{k}$ denotes the count of samples which relates to class $c_{j_{1}}$ and is classified accurately as $c_{j_{1}}$ by voter *f_k_*. When $j_{1}\neq j_{2}$, $n_{j_{1}j_{2}}^{k}$ represents the number of samples belonging to class $c_{j_{1}}$ that are misclassified as $c_{j_{2}}$ by voter *f_k_*. The instances classified as $c_{j_{2}}$ become
(12)$$n_{j_{2}}^{k}=\overset{N}{\underset{j_{1}=1}{\sum }}n_{j_{1}j_{2}}^{k}\;\left(j_{2}=1,2,\ldots ,N\right)$$

Then, the conditional probability of this sample truly belonging to class $c_{j_{1}}$ is represented as
(13)$$PM_{k}=\left[\begin{array}{cccc} P_{11}^{k} & P_{12}^{k} & \cdots & P_{1N}^{k}\\ P_{21}^{k} & \ddots & \cdots & P_{2N}^{k}\\ \vdots & \vdots & P_{j_{1}j_{2}}^{k} & \vdots \\ P_{N1}^{k} & P_{N2}^{k} & \cdots & P_{NN}^{k} \end{array}\right]\;\left(k=1,\;2,\;\ldots \;,K\right).$$
when classifier $f_{k}$ classifies instance $x_{i}$ as class $c_{j_{2}}$, its voting weight for class $c_{j_{1}}$ is $P_{j_{1}j_{2}}^{k}$.

#### 3.3.3. Model Calibration

Neural networks (NNs) are often poorly calibrated [34], i.e., they are overconfident in their predictions. In the classification process, NNs produce “confidence” scores along with the predictions. These confidence levels ideally coincide with the actual likelihood of correctness. For instance, if we provide 100 predictions with a confidence level of 80%, we expect that 80% of the predictions will be true, thus confirming that the network is calibrated. Model calibration is the process of obtaining a trained model and applying a post-processing procedure to enhance its probability estimation.

Let input images X∈x and class labels Y∈Y={1,…,k} be random variables following the joint ground-truth distribution π(X,Y)=π(Y|X)π(X); let h be a CNN with h(X)=(Y^,P^), where Y^ is the predicted class and P^∈[0,1] is the attributed confidence level. The objective is to calibrate P^ such that it represents the true class probability. In practice, the accuracy of deep learning networks is lower than its confidence. Perfect calibration is defined as
(14)ℙ(Y^=Y|P^=p)=p, ∀p∈[0,1]

The calibration error which describes the deviation in expectation between confidence and accuracy becomes
(15)$$E_{\hat{p}}=\left[\left|\left(\hat{Y}=Y|\hat{P}=p\right)-p\right|\right]$$

Calibration techniques for classifiers are aimed at converting an uncalibrated classifier’s confidence score to a calibrated score $\hat{Q}\in \left[0,1\right]$, which corresponds to the precision for a specific level of confidence [35]. This calibration technique is a post-processing technique that requires a separate learning phase to establish a mapping $g:\hat{P}\to \hat{Q}$ along with $\hat{\theta }$, which denotes the calibration parameters and can be considered as a probabilistic model $\hat{\pi }(Y|\hat{P},\hat{\theta })$. The calibration parameters $\hat{\theta }$ are typically estimated using maximum likelihood (ML) for all scaling methods while minimizing the NLL loss. Here, the calibration parameters $\hat{\theta }$ can be calculated by applying an uninformative Gaussian prior π(θ) with a wide variance over the parameters and inferring the posterior by
(16)$$\pi \left(\theta |\hat{P},Y\right)=\frac{\pi \left(Y|\hat{P},\theta \right)\pi \left(\theta \right)}{\int_{\Theta }\pi \left(Y|\hat{P},\theta \right)\pi \left(\theta \right)d\theta },$$
where $\pi \left(Y|\hat{P},\theta \right)$ is the likelihood. We can map a new input ${\hat{p}}^{*}$ with the posterior predictive distribution defined by
(17)$$f\left(y^{*}|{\hat{p}}^{*},\hat{P},Y\right)=\underset{\Theta }{\int }\pi \left(y^{*}|{\hat{p}}^{*},\theta \right)\pi \left(\theta |\hat{P},Y\right)d\theta$$

We modeled the epistemic uncertainty of calibration mapping compared to Bayesian NNs. The distribution $f_{i}$ acquired by calibration for an instance with index i reflects the uncertainty of the model for a specific degree of confidence rather than the model uncertainty for a particular prediction. A distribution is obtained as a calibrated estimate. We utilized stochastic variational inference (SVI) for estimation because the posterior cannot be calculated analytically. The SVI uses a variational distribution (often a Gaussian distribution) whose structure is simple to evaluate [35]. We sampled T sets of weights and used them to construct a sample distribution containing T estimates for a new single input ${\hat{p}}^{*}$, with the parameters of the variational distribution optimized to match the true posterior using the evidence lower bound loss.

We adopted the matrix and vector scaling method, which is a multiclass extension of Platt scaling [36]. Let $z_{i}$ be the logit vector produced before the softmax layer for input $X_{i}$. The matrix scaling applies the linear transformation $Wz_{i}+b$ to the following logits:(18)$${\hat{q}}_{i}=\underset{k}{\max}\sigma_{SM}{\left(Wz_{i}+b\right)}^{\left(k\right)},$$
(19)$${{\hat{y}}^{\prime }}_{i}=\underset{k}{\mathrm{argmax}}\sigma_{SM}{\left(Wz_{i}+b\right)}^{\left(k\right)}$$

The parameters W and b were optimized in accordance with the NLL on the validation data set. Vector scaling can be described as a variant where W is constrained to a diagonal matrix, and the number of parameters for matrix scaling accumulates quadratically along with the number of classes K.

##### Calibration Evaluation

In a recent study [37], the expected calibration error (ECE) and the maximum calibration error (MCE) were used as voting weights for basic classifiers wherein the prediction error of the ensemble model declined according to the probability calibration and performed better than simple averaging. The ECE and MCE are commonly used criteria for measuring NN calibration errors [34]. Let $B_{m}$ be the set of indices of samples whose predicted confidence falls into the interval $I_{m}=\left(\frac{m-1}{M},\frac{m}{M}\right]$, m∈M. The accuracy of $B_{m}$ is
(20)$$acc\left(B_{m}\right)=\frac{1}{\left|B_{m}\right|}\sum_{i\in B_{m}}𝟙\left({\hat{y}}_{i}=y_{i}\right),$$
where ${\hat{y}}_{i}$ and $y_{i}$ are the true and predicted labels, respectively, for sample i. The average predicted confidence of bin $B_{m}$ can be defined as
(21)$$conf\left(B_{m}\right)=\frac{1}{\left|B_{m}\right|}\sum_{i\in B_{m}}{\hat{p}}_{i},$$
where ${\hat{p}}_{i}$ is the confidence of sample i.

The ECE takes the weighted average of the bins’ accuracy/confidence differences of n number of samples:(22)$$CE=\overset{M}{\underset{m=1}{\sum }}\frac{\left|B_{m}\right|}{n}\left|acc\left(B_{m}\right)-conf\left(B_{m}\right)\right|$$

The MCE [34] primarily focuses on high-risk applications, where the maximum accuracy/confidence difference is more important than just the average, which represents the worst-case scenario:(23)$$MCE=\underset{m\in \left\{1,\ldots ,m\right\}}{\max}\left|acc\left(B_{m}\right)-conf\left(B_{m}\right)\right|$$

##### Uncertainty Evaluation

Generally, the tolerance should be significantly less for COVID-19 detection. The predictive posterior distribution shows whether a network has a high or low confidence in its choice based on the input CXR images. Epistemic and aleatoric uncertainty are two distinct types of uncertainties that contribute to predictive uncertainty in deep learning [38]. Aleatoric uncertainty considers the noise that is already present in the observations owing to class overlap and label, homoscedastic, and heteroscedastic noises, which cannot be easily eliminated even with more data. This may be because of the sensor noise in CXR imaging from the random distribution of photons during scan capture. Epistemic or model uncertainty, which does not consider all facets of the data or the lack of training data, compensates for uncertainty in the model parameters. As the amount of training data increased, the epistemic uncertainty related to the model decreased.

Prediction interval coverage probability (PICP) is a metric used for Bayesian models aimed at determining the quality of the uncertainty estimates, i.e., the probability that a sample’s true value is contained within the predictive interval. The mean prediction interval width (MPIW) is another metric used to measure the mean width of all prediction intervals (PIs) in order to evaluate the sharpness of uncertainty estimates.

A PI around the mean estimate can be used to express epistemic uncertainty. We obtained the interval boundaries by selecting quantile-based interval boundaries for a certain confidence level τ and by assuming a normal distribution:(24)Cτ,i=(li,ui): ℙ(li≤prec(i)≤ui)=1−τ,
where prec(i) represents the observed precision of sample i for a specific p^i. If all samples of the measured accuracies lie within a 100(1 − τ)% PI around 100(1 − τ)% of the time, the uncertainty is properly calibrated [35], and the use of g as a calibration model produces a PDF fi for an input with index i out of N samples. We can calculate the PICP from (25):(25)PICP=1N∑i=1N 𝟙(prec(i)∈ Cτ,i). 

The PICP is often used to perform calibrated regression when the true target value is known. However, the true precision of the classification cannot be easily obtained. Thus, we applied a binning method to all available quantities $\hat{P}$ with N points to estimate the accuracy for each instance. It is necessary for PICP →(1−τ) as N → ∞ for flawless uncertainty calibration. By using this concept, we can calculate the uncertainty by measuring the difference between PICP and (1−τ). The PI width for a certain $C_{\tau ,i}$ is averaged over the entire N data points to create the MPIW, which is a complementary metric. Regarding the two metrics, it is desirable that the models possess larger PICP values and that the MPIW is reducing. By utilizing PICP and MPIW, we can assess both the quality of the calibration mapping and the epistemic uncertainty quantification. In our study, we suggest using PICP as an ensemble voting weight.

### 3.4. Optimal Voting Weights

Despite achieving significantly high accuracy in numerous detection and classification problems, the CNN has been considered a “Black box” in recent years. Although the CNNs’ architecture, processes, and features extraction methodology are extensively studied, it is still difficult for humans to know how the network decides its classification and the selection criteria of the features, based on which the decision is made. This is especially important in sectors such as the military and medicine, where the reasoning behind the decision is important. Explainable AI research has grown significantly owing to the advancements in deep learning and the need for reliable machine decisions. The Grad-CAMs have been used extensively in medical imaging [39]. To highlight the crucial areas that are class-discriminating saliency maps, a Grad-CAM evaluates the gradients of the feature maps in the final convolution layer on a CNN model for a target image. To determine the target class weights of each filter in Grad-CAM, the gradients flowing back to the final convolutional layer in the CNN model were globally averaged. A combination of weighted feature maps and ReLU activation constitutes a Grad-CAM heat map. The expression of the class-discriminative saliency map $L^{c}$ for the target image class c is
(26)$$L_{i,j}^{c}=ReLU(\sum_{k}w_{k}^{c}A_{i,j}^{k}),\;$$
where the activation map for the k-th filter at the spatial position (i,j) is denoted as $A_{i,j}^{k}$, and ReLU highlights the favorable characteristics of the target class. The k-th filter’s target class weights (importance weights) are calculated as follows:(27)$$w_{k}^{c}=\frac{1}{Z}\sum_{i}\sum_{j}\frac{\partial Y^{c}}{\partial A_{i,j}^{k}},$$
where $Y^{c}$ is the probability of classifying the target category as c, and the number of pixels in the activation map is represented as Z. In this study, we suggest the use of the quality of epistemic uncertainty quantification measured by PICP to improve the Grad-CAM output. By integrating the PICP, the target class weights of the k -th filter become
(28)$$w_{k}^{c}=\frac{1}{Z}\sum_{i}\sum_{j}\frac{\partial (Y^{c}*\frac{1}{PICP})}{\partial A_{i,j}^{k}}.$$

The class-discriminative saliency map $L^{c}$ based on the uncertainty of the target image class c becomes
(29)$$L_{Uncertain-CAM}^{c}=[ReLU\left(\underset{k}{\sum }w_{k}^{c}A^{k})*\frac{1}{PICP}\right]$$

### 3.5. Uncertain-CAM Evaluation Metrics

For a statistical evaluation of the proposed method, we applied the intersection over union (IoU) method, also known as the Jaccard similarity coefficient, to compare the Uncertain-CAM output with the ground truth, as illustrated in Figure 2 and Algorithm 2. The IoU is a statistical method used to gauge the similarity and diversity and is commonly used as an evaluation metric for semantic image segmentation. It ranges between 0 and 1, where 0 represents the absolute difference and 1 represents the absolute match. Let $I_{\mathrm{Uncertain}-\mathrm{CAM}\;}:u\;\to {\left[0,\;1\right]}^{3}$ be the output RGB image of Uncertain-CAM mapped onto the JET color map and $I_{\mathrm{Ground}-\mathrm{Truth}}^{binary\;mask}:u\;\to {\left[0,\;1\right]}^{1}$ is the binary mask ground-truth image. We first converted $I_{\mathrm{Uncertain}-\mathrm{CAM}\;}$ to HSV color space (hue, saturation, value). The hue channel holds great significance in image processing applications that require object segmentation based on color because it represents the type of color. The saturation varies from unsaturated to representing grayscale, which is totally saturated. The value channel describes color brightness or intensity. Because the JET color map represents the feature importance to the class as it changes from blue (not relevant) to red (most relevant), we can segment $I_{\mathrm{Uncertain}-\mathrm{CAM}\;}$ using certain lower (L) and upper (U) color ranges [L, U] representing only the activated features that produce a new mask image, containing only the learned features $I_{\mathrm{Uncertain}-\mathrm{CAM}}^{binary\;mask}$, where the IoU can be applied. These can be computed as
(30)$$IoU=\frac{Area\;of\;overlap\;}{Area\;of\;Union}=\frac{I_{\mathrm{Uncertain}-\mathrm{CAM}}^{binary\;mask}\cap^{\;}I_{\mathrm{Ground}-\mathrm{Truth}}^{binary\;mask}}{I_{\mathrm{Uncertain}-\mathrm{CAM}}^{binary\;mask}\cup^{\;}I_{\mathrm{Ground}-\mathrm{Truth}}^{binary\;mask}}$$

**Algorithm 2:** Uncertain-CAM evaluation algorithm.Input: IUncertain−CAM :u →[0, 1]3; IGround−Truthbinary mask:u →[0, 1]1Output: IoUinitialization;for i=1,…,IUncertain−CAM  do**1:** Read IUncertain−CAM :u →[0, 1]3**2:** Convert IUncertain−CAM  to HSV Colorspace. **3:** Set Color range boundaries [L, U]
**4:** Create new binary mask IUncertain−CAMbinary mask**5:** Compute IoU using (30)**end**

## 4. Results

### 4.1. Data Processing

Data preprocessing is a crucial stage of data cleaning that enhances input data in deep learning projects. It guarantees the generalizability of a particular model, particularly when tested on datasets distinct from those used for training. Preprocessing further improves network performance by minimizing data noise and/or distortions. In our tests, we first used data normalization, rescaling all pixel values to [0, 1] using a pixel-wise multiplication factor of 1/255, yielding a set of CXR images as follows:(31)$$N_{i}=\frac{I-I_{min}}{I_{max}-I_{min}},$$
where $I_{max}\;$and $I_{min}\;$are the maximum and lowest values in the input data, respectively, and $N_{i}$ is the normalized data. The histogram equalization was used to improve the image contrast [40]. The images were then downsized to 224 × 224 pixels before the training started. Additionally, the data were adjusted by a 25° rotation, zoom range of 0.2, and alteration of the fill-mode to the nearest. Because data augmentation reduces overfitting, which usually occurs when a statistical model exactly fits its training data, it was used to increase the generalization capacity of our model by introducing changes to the dataset. Unfortunately, when this occurs, the goal of the algorithm is defeated because it cannot perform accurately against unobserved data [41].

The variations between a few COVID-19 pneumonia case images are shown in Figure 3. In the CXRs of COVID-19 patients, the following principal features are typically observed [42]:GGOs.Odd paving pattern.Consolidation of the airspace.Thickening of bronchovascular bundles.Traction bronchiectasis.

Similarly, the following CXR features have been observed in pneumonia patients [43]:GGOs.Reticular opacities.Vascular thickness.Additional widespread distribution along the bronchovascular bundles.Thickness in bronchial wall.

Although these features are often seen in pneumonia, isolated lobar, or segmental consolidation without GGO, numerous small pulmonary nodules, tree-in-bud, pneumothorax, cavitation, and hilar lymphadenopathy are uncommon in COVID-19 cases [42]. In addition to the procedures used for the early diagnosis, cure, and isolation phases, radiological imaging is crucial in stemming the outbreak. Chest radiography can be used to identify a few distinctive lung abnormalities associated with COVID-19 infection. Conversely, deep learning algorithms are adept at spotting COVID-19 lung symptoms, and they yield excellent diagnostic accuracy rates.

### 4.2. Learning COVID

Both the COVID-19 classification and the explanation of the suggested approach were subjected to quantitative evaluation. The ResNet50 [44], Inception V3 [45], and the VGG16 [46] were the foundation models employed in this study. The VGG16 CNN architecture won the 2014 ILSVR (ImageNet) competition and is regarded one of the best visual model architectures created to date. The most distinctive feature of VGG16 is that it emphasizes having convolution layers of 3 × 3 filters with a stride of 1, and it always utilizes the same padding and maxpool layer of 2 × 2 filters with a stride of 2. Throughout the architecture, the convolution and max pool layers were arranged in the same manner. It ends with two fully connected layers and a softmax layer for the output. The 16 layers in VGG16 indicate that there are 16 layers with weights. This network has 138.4 million parameters, making it sizable. The ResNet50 CNN comprises 48 convolutional layers, a maxpool layer, and an average pool layer. The framework introduced by ResNets made it possible to train extremely deep-NN, i.e., the network may have hundreds or thousands of layers and still function well. In the ImageNet dataset, the image recognition model Inception v3 achieved an accuracy of approximately 78.1%. This model has incorporated numerous concepts that are established by various researchers over the years. Convolutions, average pooling, max pooling, concatenations, dropouts, and fully linked layers are symmetric and asymmetric building components that constitute the model itself. The computational efficiency of inception networks has been demonstrated in terms of the number of parameters created by the network and economic cost (memory and other resources) [45].

We used 3-fold cross-validation to realize the model formulation and evaluation processes. Three snapshot models were used for each fold cross-validation, resulting in nine models for the three folds. The data were split into three equivalent subsets that were mutually exclusive for 3-fold cross-validation. Two of the three subsets were utilized as the training set in each iteration, and the third subset was used as the validation set. The final evaluation of each model depended on the average ensemble performance of the nine snapshots created for each model. Before the final weighted voting ensemble stage, each base model was calibrated in order to obtain base model performance, calibration errors, model confidence, and uncertainty. All experiments were repeated ten times in order to minimize statistical variability. 

### 4.3. Performance Evaluation Metrics

During classification training, the evaluation metric is crucial for obtaining the best classifier. Therefore, choosing an appropriate assessment measure is important to differentiate and achieve the best classifier [47]. We considered the following evaluation measures to statistically assess the efficacy of our suggested method: area under the curve (AUC), a popular ranking metric; accuracy, which evaluates the proportion of accurate predictions over the entire count of samples evaluated; precision, which evaluates the proportion of positive patterns that are correctly classified; recall, for evaluating the proportion of positive patterns that are accurately predicted; and F1-score, which describes the harmonic mean between recall and precision values [47]. Let TP and TN represent true positive and true negative, and FP and FN represent false positive and false negative, respectively. Then, the performance metrics are calculated as follows:(32)$$ACC=\frac{\left(TP+TN\right)\;}{Total\;samples},$$
(33)$$Precison=\frac{TP}{TP+FP},$$
(34)$$Recall=\frac{TP}{TP+FN},$$
(35)$$F1=\frac{2TP\;}{\left(2TP+FP+FN\right)}.$$

### 4.4. Performance Evaluation

In this section, we statistically evaluated the proposed method on an unseen (test) dataset. Table 1 and Table 2 show both calibration results and performance of the voters. Table 3 and Table 4 list the calibration results and performance of the ensemble models based on different voting strategies. Table 5 presents the performance results for each class for both the base classifiers and ensemble models. The smaller the calibration errors ECE and MCE, the better. For uncertainty quality metrics, we aimed to maximize PICP and minimize MPIW. A few errors can be seen before calibration, and they highlight the overconfident issue of CNN. For unseen data, the ensembled models overperform individual models, and the proposed weighted voting ensembled outperforms other voting strategies and provides a better generalization on unseen data.

Figure 4 shows the performance of the proposed network for the unseen data and compares the models at different stages of proposed training scheme. The ensembled snapshots overperform the simple cross-validation and the PICP-based ensemble produces a more efficient performance.

### 4.5. Explaining COVID-19

Deep-CNNs are described as “black boxes” because they are not naturally interpretable. Accordingly, a global statement on the ethics of AI in radiology, transparency, interpretability, and explainability is required to establish trust between the patient and the provider. Any researcher or physician that deploys such a system should understand, via reading the explanations, how the model reached a specific conclusion regarding the presence of COVID-19 infection or absence thereof. It is not only morally right to seek out explainable machine learning models but also instructive for researchers to confirm that the model contains no data leaks or unintentional bias.

Table 6 provides some samples of explanations for each model that made positive predictions of true positive COVID-19 CXR as heatmaps using the Grad-CAM, where dark red regions represent the most relevant information that the model depends on when making its decision, blue represents the least significant importance, and the ground truth represents the marking of specialized physicians. The proposed Grad-CAM ensemble combines the average weights of each basic model, and the proposed Uncertain-CAM applies uncertainties as weights for each basic model. Figure 5 provides a comparison of the IoU score between the ground truth and the explanation done directly to single models, ensemble model, and the Uncertain-CAM. It demonstrates that our proposed method generates a higher score than the single model explanation; higher scores mean more closeness to the ground truth, as determined by specialized physicians.

## 5. Discussion

Recently, COVID-19 detection using X-rays has been extensively studied by researchers across the world. Although various approaches have achieved outstanding results in medical imaging tasks, a majority of these models suffer from high epistemic uncertainty because the small amounts of available data are mostly overconfident and have not been deployed because of lack of reliability, risk of erroneous decisions, and poor generalization on unseen data [48]. In our study, the performance and feature extraction capability of the base models vary, which further demonstrates the efficiency of our proposed network structure to learn better. Table 7 shows a comparison of the effectiveness of our strategy with that of the strategies that are currently employed. Our approach exhibited state-of-the-art performance for both classification and explanation compared to other studies for the same dataset [62,63]. This superior performance may be attributed to the high-quality and large 33,000 CXR images, which could be unfair to compare with other studies with different datasets. Ensemble learning proved to be more reliable not only in terms of accuracy but also in reducing CNN overconfidence. Although CNN explainability still has many shortcomings, the ensemble approach demonstrated great potential to produce a more accurate and meaningful visualization of what was really learned. The calibration and uncertainty not only improve the ensemble prediction but also provide a more reliable output.

While the results are encouraging, our analysis pipeline has certain limitations. First, we used a single benchmark dataset to train and test our proposed method. To improve the aleatoric uncertainty evaluation and further strengthen the robustness of our technique, it is ideal to leverage external test (unseen) datasets from various centers. Second, only the CXR images were used as inputs in our methodology. Multiple data sources are frequently employed in clinical practice, including clinical biomarkers and occasional chest CT scans, because they can offer more granular views of damaged lungs. Therefore, integrating multiple inputs can provide a more robust prediction at the patient level. Third, our system utilizes three standard base models, and the investigation of the number and structure of different models should provide more insights on both the accuracy and explanation of our proposed system. Finally, the use of deep learning in medical fields has great potential but is still obscure, as we are yet to fully explain what the machine really learns, wherein the need for reliable explaining tools is indispensable.

## 6. Conclusions

Previous studies on computational medical imaging focused on investigating the possibilities of CNN applicability in medical images produced by CT, MRI, and X-rays because of the significant advancements due to the CNN technology. Existing methodologies primarily concentrate on achieving high-performance accuracy without adequately elaborating on what CNNs actually learn or delving into fundamental CNN problems, including the overconfidence predictions and uncertainty evaluation, which are crucial factors in the field of medical diagnosis. In this study, we propose a novel weighted average ensemble method based on uncertainty to combine the confidence scores of various CNN models and subsequently maximize their efficiency for both COVID-19 classification and explanation. The proposed methodology uses uncertainty to provide CNN predictions and judgment explanations that are more robust and trustworthy by presenting both high classification accuracy and more insightful justification. The results demonstrated the capacity of the proposed framework to correctly detect COVID-19 cases in a cohort of 33,000 CXR images. The proposed approach outperforms existing deep-learning-based systems for COVID-19 classification in terms of accuracy and trustworthiness. Further, it can be widely used in clinical applications and help front-line medical staff diagnose COVID accurately and quickly. Future research will expand the capacity of our system to combine several inputs, such as clinical indicators and other imaging modalities (such as chest CT), to produce more thorough markers (clinical and visual) that can aid doctors in providing more effective and customized treatment. Furthermore, we will attempt to test our model using a variety of datasets in future studies.

## Figures and Tables

**Figure 1 diagnostics-13-00441-f001:**
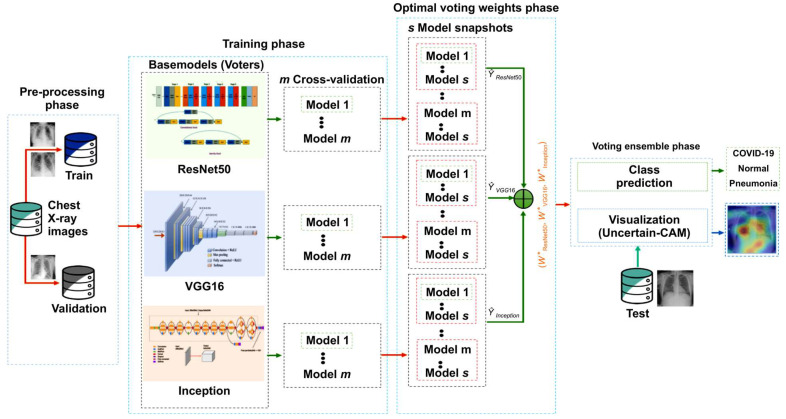
Framework of the proposed method. In the preprocessing stage, data were prepared for training phase wherein we generate m models using cross-validation for each CNN network, and for each model, we generate s snapshots models using cyclic cosine annealing. Snapshots models were used to obtain optimal voting weight.

**Figure 2 diagnostics-13-00441-f002:**
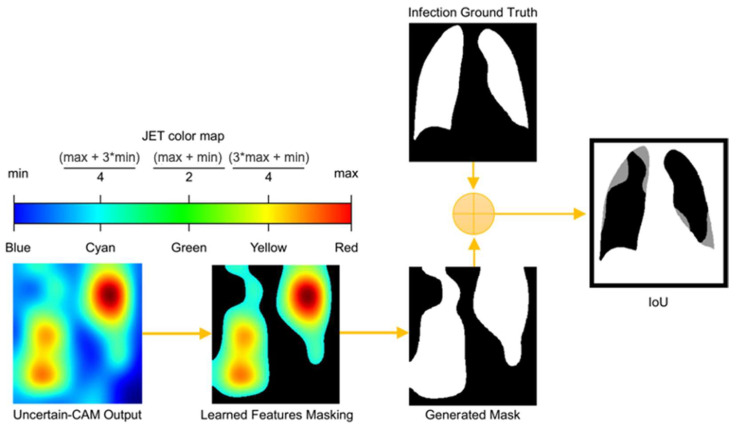
Process of computing IoU from Uncertain-CAM Output and Ground Truth. The generated heatmap is masked based on color threshold and is compared with ground truth masked to generate IoU, which describes how much generated masked is aligned with the ground truth masked.

**Figure 3 diagnostics-13-00441-f003:**
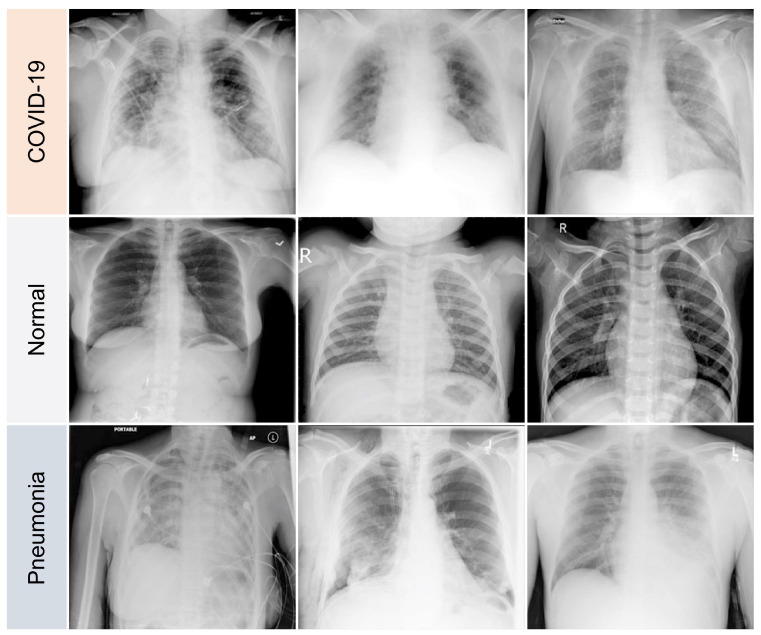
Samples of dataset used in the study.

**Figure 4 diagnostics-13-00441-f004:**
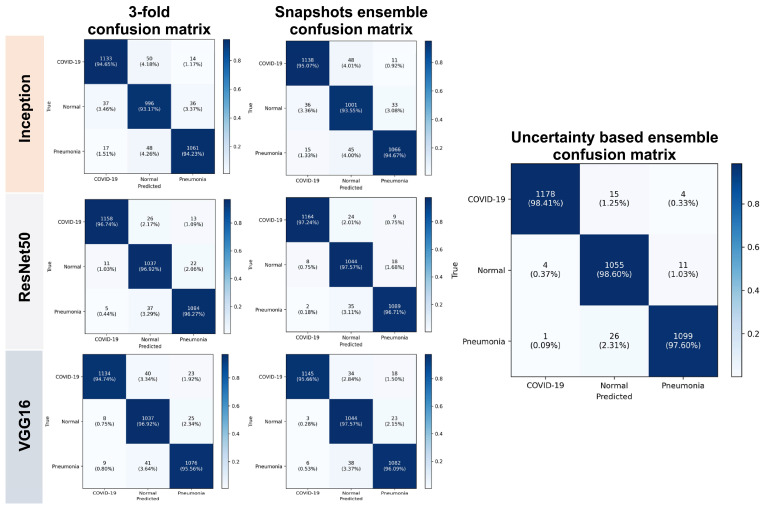
Proposed network performance output on unseen data.

**Figure 5 diagnostics-13-00441-f005:**
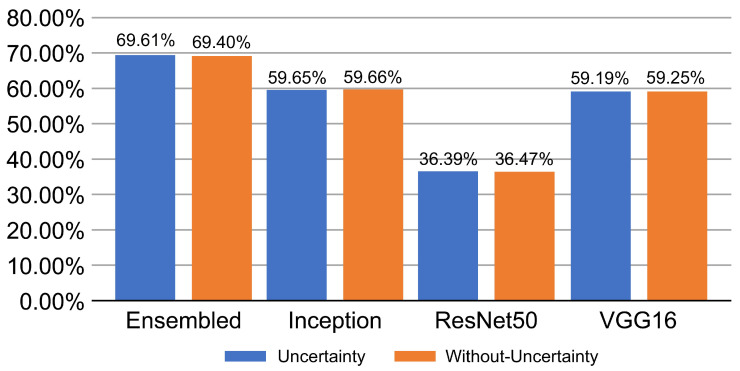
IoU scores with and without uncertainty.

**Table 1 diagnostics-13-00441-t001:** Calibration (in %) of voters on unseen data.

Voters	ECE-B	ECE-A	MCE-B	MCE-A	PICP	MPIW
VGG16	61.83 (61.47–62.90)	0.83 (0.28–0.96)	64.60 (64.51–65.01)	**21.16** (20.72–21.39)	92.57 (92.20–92.91)	4.74 (4.33–4.97)
ResNet50	61.93 (61.76–62.30)	**0.48** (0.09–0.45)	64.55 (64.12–64.67)	47.86 (47.64–48.18)	90.66 (89.97–91.14)	5.96 (5.31–6.20)
Inception	**56.63** (56.63–56.83)	1.09 (1.07–1.57)	**63.52** (63.16–63.70)	37.77 (37.48–38.13)	**90.83** (90.81–91.61)	6.85 (6.38–6.91)

In each cell M (P25–P75), M is the median, P25 is the 25th percentile, and P75 is the 75th percentile of 10 tests. For each model, the best performance of each column is indicated in bold. A indicates ”after calibration,” B indicates “before calibration.”.

**Table 2 diagnostics-13-00441-t002:** Performance (in %) of voters on unseen data.

Voters	Precision	Recall	F1	ACC	AUC
VGG16	96.37 (96.01–96.95)	96.44 (95.95–96.69)	96.38 (96.25–96.93)	96.40 (96.27–96.68)	97.33 (96.60–97.41)
ResNet50	**97.13** (96.84–97.47)	**97.18** (96.72–97.64)	**97.14** (96.95–97.52)	**97.17** (96.67–97.43)	**97.89** (97.38–98.10)
Inception	94.42 (94.35–94.75)	94.43 (94.37–94.83)	94.42 (94.19–94.65)	94.46 (94.46–94.95)	95.84 (95.45–95.82)

In each cell M (P25–P75), M is the median, P25 is the 25th percentile, and P75 is the 75th percentile of 10 tests. For each model, the best performance of each column is indicated in bold. A indicates ”after calibration”, and B indicates “before calibration”.

**Table 3 diagnostics-13-00441-t003:** Calibration (in %) of the ensembled models on unseen data.

Strategy	ECE-B	ECE-A	MCE-B	MCE-A	PICP	MPIW
Majority Voting	58.88 (58.89–59.23)	0.65 (0.62–0.69)	62.55 (62.41–62.78)	53.92 (53.47–53.93)	72.86 (63.6–81.17)	7.96 (4.6–10.69)
Best Combination	58.48 (58.02–58.78)	0.73 (0.70–0.76)	**62.36** (61.92–62.54)	21.44 (20.70–22.08)	68.43 (62.81–74.21)	4.59 (2.33–5.91)
Priori Recognition Performance	**54.82** (54.79–54.85)	**0.24** (0.22–0.27)	89.45 (89.40–89.46)	42.33 (42.29–42.37)	37.61 (36.78–41.71)	5.55 (2.32–5.33)
ECE (ours)	98.35 (98.32–98.38)	0.86 (0.85–0.88)	62.47 (62.41–62.52)	53.85 (53.8–53.89)	70.51 (66.52–73.78)	**3.66** (1.72–6.14)
MCE (ours)	58.96 (58.9–58.98)	0.58 (0.51–0.62)	62.69 (62.64–62.77)	**13.79** (13.71–13.89)	64.67 (63.29–67.48)	4.76 (2.76–6.86)
PICP (ours)	59.15 (58.7–59.54)	0.60 (0.43–0.81)	63.00 (62.62–63.49)	19.57 (19.03–19.81)	**77.25** (69.25–82.19)	4.16 (1.52–6.63)

In each cell M (P25–P75), M is the median, P25 is the 25th percentile, and P75 is the 75th percentile of 10 tests. For each model, the best performance of each column is indicated in bold. A indicates ”after calibration” and B indicates “before calibration”.

**Table 4 diagnostics-13-00441-t004:** Performance (in %) of ensembled models on unseen data.

Strategy	Precision	Recall	ACC	F1	MCC	AUC
Majority Voting	97.71 (97.27–98.09)	97.75 (97.57–97.97)	97.76 (97.37–98.30)	97.73 (97.46–98.14)	98.32 (98.10–98.57)	97.71 (97.27–98.09)
Best Combination	97.87 (97.47–98.03)	97.95 (97.92–97.96)	97.95 (97.93–97.97)	97.92 (97.88–97.97)	98.47 (98.44–98.50)	97.87 (97.47–98.03)
Priori Recognition Performance	97.70 (97.66–97.74)	97.77 (97.71–97.81)	97.74 (97.68–97.79)	97.75 (97.73–97.78)	98.31 (98.27–98.36)	97.70 (97.66–97.74)
ECE (ours)	98.11 (98.04–98.14)	**98.18** (98.14_98.2)	98.20 (98.16–98.24)	98.12 (98.07–98.17)	98.67 (98.66–98.73)	98.11 (98.04–98.14)
MCE (ours)	98.15 (98.09–98.2)	**98.18** (98.12–98.2)	98.19 (98.16–98.21)	98.13 (98.09–98.16)	98.63 (98.58–98.66)	98.15 (98.09–98.2)
PICP (ours)	**98.18** (98.12–98.23)	98.17 (98.11–98.22)	**98.24** (98.2–98.27)	**98.20** (98.13–98.29)	**98.71** (98.66–98.76)	**98.18** (98.12–98.23)

In each cell M (P25–P75), M is the median, P25 is the 25th percentile, and P75 is the 75th percentile of 10 tests. For each model, the best performance of each column is indicated in bold. A indicates ”after calibration” and B indicates “before calibration”.

**Table 5 diagnostics-13-00441-t005:** Performance of the ensembled models per class.

Strategy	Class	Precision	Recall	ACC	F1	AUC
VGG16	COVID-19	95.63 (95.54–95.69)	95.68 (95.59–95.76)	98.21 (98.18–98.23)	97.36 (97.31–97.43)	97.59 (97.55–97.63)
	Pneumonia	96.36 (96.33–96.41)	96.07 96.03–96.12)	97.47 (97.4–97.54)	96.24 (96.18–96.31)	97.17 (97.15–97.23)
	Normal	93.52 (93.46–93.57)	97.62 (97.55–97.66)	97.13 (97.1–97.17)	95.54 (95.51–95.6)	97.21 (97.13–97.29)
ResNet50	COVID-19	99.13 (99.07–99.2)	97.22 (97.19–97.27)	98.73 (98.65–98.78)	98.19 (98.12–98.24)	98.44 (98.42–98.5)
	Pneumonia	97.57 (97.47–97.64)	96.70 (96.64–96.75)	98.13 (98.04–98.21)	97.17 (97.13–97.23)	97.74 (97.69–97.79)
	Normal	94.65 (94.6–94.67)	97.54 (97.47–97.63)	97.53 (97.47–97.56)	96.05 (95.97–96.1)	97.55 (97.5–97.61)
Inception	COVID-19	95.72 (95.67–95.81)	95.03 (94.96–95.11)	96.78 (96.7–96.87)	95.47 (95.4–95.5)	96.32 (96.26–96.37)
	Pneumonia	96.01 (95.95–96.07)	94.68 (94.64–94.73)	96.94 (96.91–97)	95.38 (95.3–95.44)	96.35 (96.32–96.38)
	Normal	91.46 (91.42–91.49)	93.53 (93.48–93.59)	95.23 (95.17–95.28)	92.55 (92.52–92.58)	94.80 (94.76–94.85)
Majority Voting	COVID-19	99.23 (99.04–99.72)	**98.70** (98.16–99.5)	99.38 (98.9–99.7)	98.60 (98.03–99.26)	99.03 (98.44–99.44)
	Pneumonia	97.81 (97.48–98.32)	97.32 (96.99–97.61)	98.50 (98–99.17)	97.57 (96.88–98.07)	98.32 (97.86–98.69)
	Normal	95.46 (94.59–96.4)	97.64 (97.2–97.76)	97.77 (97.28–98.52)	97.26 (97.06–97.62)	97.97 (97.08–98.89)
Best Combination	COVID-19	99.49 (99.45–99.54)	97.82 (97.8–97.86)	98.99 (98.87–99.46)	98.65 (98.58–98.72)	98.60 (98.1–99.22)
	Pneumonia	98.45 (98.45–98.47)	97.91 (97.64–98.34)	98.70 (98.07–99.33)	98.12 (98–98.89)	98.27 (97.8–98.7)
	Normal	95.98 (95.42–96.49)	**98.87** (98.36–99.38)	98.08 (97.74–98.57)	97.19 (96.56–97.84)	97.77 (97.47–98.24)
Priori Recognition Performance	COVID-19	98.91 (97.61–100.02)	98.14 (97.47–98.75)	99.16 (98.95–99.42)	98.90 (98.73–99.18)	98.92 (98.75–99.14)
	Pneumonia	98.07 (97.77–98.77)	97.00 (96.89–97.17)	98.54 (98.32–98.63)	97.45 (97.18–97.81)	98.08 (97.89–98.25)
	Normal	95.70 (95.29–96)	98.00 (97.76–98.24)	97.98 (97.49–98.39)	96.74 (96.55–97.16)	97.90 (97.81–98.15)
ECE (ours)	COVID-19	99.53 (99.32–99.74)	98.12 (97.83–98.21)	99.17 (98.88–99.42)	98.81 (98.44–99.01)	99.04 (98.74–99.49)
	Pneumonia	98.32 (98.25–98.39)	97.57 (97.3–97.83)	98.75 (98.44–99.08)	98.02 (97.87–98.12)	98.44 (98.32–98.69)
	Normal	96.16 (95.96–96.32)	98.57 (98.09–98.96)	98.30 (98.17–98.49)	97.37 (97.22–97.72)	98.39 (97.91–98.73)
MCE (ours)	COVID-19	99.52 (99.31–99.73)	98.46 (98.04–98.77)	99.32 (99.15–99.58)	99.01 (98.58–99.42)	99.11 (98.84–99.27)
	Pneumonia	98.66 (98.48–98.83)	97.41 (97.19–97.66)	98.68 (98.38–98.93)	98.02 (97.79–98.3)	98.37 (98.13–98.7)
	Normal	96.15 (95.96–96.25)	98.59 (98.35–98.75)	98.32 (98.17–98.45)	97.35 (97.09–97.65)	98.41 (98.08–98.82)
PICP (ours)	COVID-19	**99.65** (99.46–99.84)	98.43 (98.2–98.75)	**99.48** (99.35–99.65)	**99.20** (98.81–99.53)	**99.27** (98.99–99.4)
	Pneumonia	**98.89** (98.68–99.14)	**97.65** (97.37–97.73)	**98.81** (98.58–98.99)	**98.15** (97.81–98.47)	**98.54** (98.43–98.59)
	Normal	**96.33** (95.96–96.65)	98.66 (98.4–99.13)	**98.47** (98.04–98.82)	**97.59** (97.26–98.03)	**98.58** (98.35–98.86)

In each cell M (P25–P75), M is the median, P25 is the 25th percentile, and P75 is the 75th percentile of 10 tests. For each model, the best performance of each column is indicated in bold. A indicates ”after calibration” and B indicates “before calibration”.

**Table 6 diagnostics-13-00441-t006:** Explainability of positive COVID-19 X-ray classified as positive.

X-ray	VGG16	ResNet50	Inception	Ensembled (Ours)	Uncertain-CAM (Ours)	Ground Truth
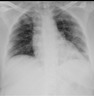	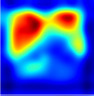	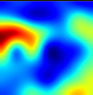	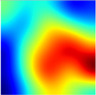	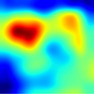	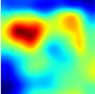	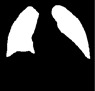
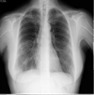	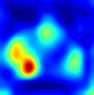	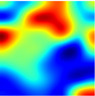	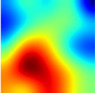	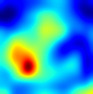	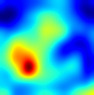	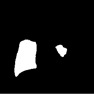
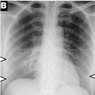	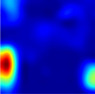	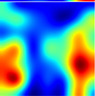	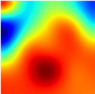	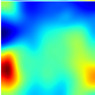	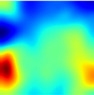	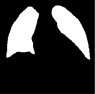

**Table 7 diagnostics-13-00441-t007:** Performance of our proposed method compared to state-of-the-art methods.

Method	Technique	ACC	Recall	Precision	F1	AUC
(Wang et al., 2020) [49]	COVID-Net	93.3	91.0	92.80	-	-
(Ozturk et al., 2020) [17]	DarkCovidNet	87.02	85.35	89.96	87.37	-
(Khan et al., 2020) [50]	CoroNet	95.0	96.9	95.0	95.60	-
(Makris et al., 2020) [51]	Deep Learning	95.88	96.0	96.0	96.0	-
(Luz et al., 2021) [52]	EfficientNet-B3	93.94	80.6	-	-	
(Chowdhury et al., 2021) [21]	Ensemble Snapshots	96.07	97.00	94.17	86.0	99.71
(Manokaran et al., 2021) [53]	DenseNet201	92.19	94.00	-	90.00	98.33
(Monshi et al., 2021) [54]	CovidXrayNet	95.82	95.43	96.93	96.16	99.29
(Pham et al., 2021) [55]	SqueezeNet	97.47	**98.48**	94.20	96.30	**99.9**
(Chaudhary et al., 2021) [22]	Ensemble Deep Learning	95.92	95.92	-	-	-
(Abdar et al., 2021) [56]	UncertaintyFuseNet	96.35	96.37	96.35	96.36	100
(Aslan et al., 2022) [57]	Deep Learning and Machine Learning	96.29	96.42	96.42	96.41	-
(Karim et al., 2022) [58]	CNN + ALO + NB	98.01	96.04	97.87	97.45	-
(Saxena et al., 2022) [59]	CNN	92.63	91.87	95.76	93.78	-
(Chakraborty et al. 2022) [20]	Transfer Learning	96.43	93.68	-	93.0	-
(Yang et al., 2022) [41]	Ensemble Deep Learning	97.75	97.95	97.55	97.75	-
(Banerjee et al., 2022) [60]	Ensemble Deep Learning	96.39	95.69	96.97	96.30	-
(Gour and Jain, 2022) [61]	UA-ConvNet	97.67	98.15	97.87	97.99	99.65
(Ibrokhimov and Youngwook Kang) [62]	Deep Learning	95.85	95.82	97.95	95.80	97.33
(Constantinou et al., 2023) [63]	Deep Learning	95.65	95.63	97.85	95.60	-
Proposed	Uncertain-CAM	**98.24**	98.17	**98.18**	**98.20**	98.71

For each method, the best performance of each column is indicated in bold. “-“ indicates missing information from the source.

## Data Availability

The data presented in this study are openly available at https://www.kaggle.com/datasets/anasmohammedtahir/covidqu (accessed on 20 February 2022).

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
