# Peer review of "Uncertain-CAM: Uncertainty-Based Ensemble Machine Voting for Improved COVID-19 CXR Classification and Explainability"

_diagnostics, 2023, doi:10.3390/diagnostics13030441_

Round 1

Reviewer 1 Report

The paper proposes a scheme for both classification and explainability of COVID-19 X-ray chest images. For this purpose, ensemble of three network architectures, namely Resnet50, VGG16 and Inception was implemented. Classification decision was taken based on the voting scheme. All these architectures along with classifier ensemble concept are already well known. Obtained classification metrices are comparable with other state-of-the-art methods (Table 5). Thus, the advantage of the proposed classification scheme is not clear. Also, explainabilty of positive Covid-19 X-ray data is not convincing. None of heat maps presented in Table 4 shoes accurately the diagnostically most important regions when compared to the ground truth images. Such maps are rather useless for physicians in determining the most salient x-ray image fragments. Thus, this paper doesn’t contribute much to field neither in X-ray image classification nor in uncertainty evaluation.
Another limitation of this work is that the image databases used to train the algorithms come from 2020 and 2021. The virus variants that caused Sars-COV-2 then was completely different than the current one. The lung lesions caused by the delta variant are different - more benign. Therefore, the presented algorithms developed for non-existing COVID-19 variants are only of historical significance and will not be able to be used in clinical practice.  

Author Response

Response to Reviewer 1 Comments

Reviewer 1 Comments to Author:

The paper proposes a scheme for both classification and explainability of COVID-19 X-ray chest images. For this purpose, ensemble of three network architectures, namely Resnet50, VGG16 and Inception was implemented. Classification decision was taken based on the voting scheme. All these architectures along with classifier ensemble concept are already well known. Obtained classification metrices are comparable with other state-of-the-art methods (Table 5). Thus, the advantage of the proposed classification scheme is not clear. Also, explainabilty of positive Covid-19 X-ray data is not convincing. None of heat maps presented in Table 4 shoes accurately the diagnostically most important regions when compared to the ground truth images. Such maps are rather useless for physicians in determining the most salient x-ray image fragments. Thus, this paper doesn’t contribute much to field neither in X-ray image classification nor in uncertainty evaluation.

Another limitation of this work is that the image databases used to train the algorithms come from 2020 and 2021. The virus variants that caused Sars-COV-2 then was completely different than the current one. The lung lesions caused by the delta variant are different - more benign. Therefore, the presented algorithms developed for non-existing COVID-19 variants are only of historical significance and will not be able to be used in clinical practice. 

Response:

We thank the reviewer for the detailed and insightful review of our manuscript. We understand the concerns raised by the reviewer regarding the drawbacks of our study, which may limit practical use. However, we believe that the findings of the manuscript are still significant, and they outweigh the drawbacks. We have emphasize on all points raised by the reviewer, and modified the manuscript accordingly, and we have provided detailed responses to the comments of the reviewer below.

Point 1: The paper proposes a scheme for both classification and explainability of COVID-19 X-ray chest images. For this purpose, ensemble of three network architectures, namely Resnet50, VGG16 and Inception was implemented. Classification decision was taken based on the voting scheme. All these architectures along with classifier ensemble concept are already well known. Obtained classification metrices are comparable with other state-of-the-art methods (Table 5). Thus, the advantage of the proposed classification scheme is not clear.

Response 1: We thank the reviewer for the insightful comments. The advantages of the proposed approach are summarized below:

  • Miscalibration is an underrated issue in CNN, and we have capitalized on this situation and converted this issue into an advantage through uncertainty quantification that is measured using PICP as ensemble voting weight.
  • Effective ensembles require a diverse set of skillful ensemble members with varying distributions of prediction errors. In our training framework, we suggested the integration of the cyclic cosine annealing approach with cross-validation to perform voting for each network for accurate predictions of each network before final voting.

The output of our training framework exhibits relatively low generalization error and improved performance compared to other frameworks.

Point 2: Explainabilty of positive Covid-19 X-ray data is not convincing. None of heat maps presented in Table 4 shoes accurately the diagnostically most important regions when compared to the ground truth images. Such maps are rather useless for physicians in determining the most salient x-ray image fragments. Thus, this paper doesn’t contribute much to field neither in X-ray image classification nor in uncertainty evaluation.

Response 2: We agree that the heat maps generated may not be ready for direct use by physicians. The explainabilty is a major issue in deep learning, and thus, we tried to address this issue by enhancing the current XAI approaches. The Grad-CAM is a widely adopted XAI approach that is used to explain the CNN related to the data. We suggest the integration of the quality of epistemic uncertainty quantification measured by PICP to improve the Grad-CAM output. We also proposed a new image processing method to calculate IoU score in order to compare our approach with normal Grad-CAM. Even though the explainabilty may not be ready for practical applications, we have still contributed to the enhancement of the CNN explainabilty.

Point 3: Another limitation of this work is that the image databases used to train the algorithms come from 2020 and 2021. The virus variants that caused Sars-COV-2 then was completely different than the current one. The lung lesions caused by the delta variant are different - more benign. Therefore, the presented algorithms developed for non-existing COVID-19 variants are only of historical significance and will not be able to be used in clinical practice. 

Response 3: We used the COVID-QU-Ex, which is the largest and most comprehensive COVID-19 database currently available. Even though the virus variants may yield different features and datasets, the proposed training and explainabilty approaches are valid, and our framework could be applied to any X-ray datasets pipelines.

Reviewer 2 Report

1- the abstract section is too short and lacks much information

2- the last paragraph in the introduction section needs to be rewritten and more details about the proposed method and the contributions should be added.

"a novel CNN training structure to maximize ensembled model performance." 

what do you mean by "a novel CNN training structure"?

3- more information about the data and how it is collected should be added in 3.1. Data Preparation

4- Add a brief introduction to section 3.3. Optimal Voting Weights to explain all steps in general before going into details.

5- In the experiments section, tables 1 and 2 contain unknown numbers for ECE-B ECE-A MCE-B MCE-A ???? ??? without explanation

6- Please compare the standard Grad-CAM and the proposed one and what the differences are since figure 4 shows similar results

7- add a discussion for every table and figure you add in the experiments. There is no discussion about the results, e.g. which one is better and why.

8- Could you compare your work with other methods that use the same dataset 

Author Response

Response to Reviewer 2 Comments

Point 1: The abstract section is too short and lacks much information

Response 1: We understand the concern raised by the reviewer. Based on the comment of the reviewer, we have added significant information in the Abstract for a clear understanding of the objectives of the research.

Point 2: the last paragraph in the introduction section needs to be rewritten and more details about the proposed method and the contributions should be added."a novel CNN training structure to maximize ensembled model performance." what do you mean by "a novel CNN training structure"?
Response 2: We apologize for the confusion in the previous manuscript. We intended to state “a novel CNN training scheme.” We have made corresponding revisions in this paragraph as well as the last paragraph in the introduction section.

Point 3: More information about the data and how it is collected should be added in 3.1. Data Preparation

Response 3: Based on the suggestion of the reviewer, we have rewritten this section and included detailed information regarding the collection of data.

Point 4:  Add a brief introduction to section 3.3. Optimal Voting Weights to explain all steps in general before going into details.

Response 4: Based on the suggestion of the reviewer, a brief introduction has been added to the revised manuscript.

Point 5: In the experiments section, tables 1 and 2 contain unknown numbers for ECE-B ECE-A MCE-B MCE-A ???? ??? without explanation.

Response 5: The tables have been extended and reformatted for a clearer understanding.

Point 6:  Please compare the standard Grad-CAM and the proposed one and what the differences are since figure 4 shows similar results.

Response 6: We suggest referring to Table 4, as Figure 4 shows the performance of the proposed models according to unseen data. Table 4 compares the heatmaps of both the standard Grad-CAM and proposed methods. The output heatmaps of the basic classifiers (voters) namely, VGG16, ResNet50 and Inception, shown in Table 4, are the standard Grad-CAM, and the proposed Ensembled heatmap is the ensembled heatmap of the basic classifiers with Standard Grad-CAM. The Uncertain-CAM represents the proposed uncertainty-based Grad-CAM.

Point 7:  add a discussion for every table and figure you add in the experiments. There is no discussion about the results, e.g. which one is better and why.

Response 7: Based on the suggestion of the reviewer, a detailed discussion has been add in Section 4.4.

Point 8 Could you compare your work with other methods that use the same dataset 
Response 8: Based on the suggestion of the reviewer, we reviewed two studies[ref: 62 & 63] that integrated the same dataset, and we inferred that the proposed approach outperformed these studies.

Reviewer 3 Report

Accepted

Author Response

Thank you for taking the time to review and accept our manuscript. Sections and language of the manuscript has been modified and edited for further improvement.

Reviewer 4 Report

1. On line 166 to 168 and line 174 to 177, the serial number of equations are not alignment. And the size of equation which is on line 174 is not same as other equations. In equation (10), the comma after the first equation is lost.

2. There is a lack of explanation of Cj*from equation (1) on line 161 to 165.

3. In equation (2), it's much clearer to put π in front of the division equation.

4. There should be more explanation of the model architecture in the title of Figure 1 to make it easier for readers to understand. The title of Figure 2 has the same problem.

5. On line 230,  can’t be found in equation (9). It should probably be checked for a case error.

6. On line 315, there is an error in the parentheses of the  equation.

7. The data of the tables in the article is too messy to highlight the key points.

In addition, the manuscript contains a number of typographical errors. For instance, The layout of equation (9) and equation (10), equation (11) and equation (13) don't line up. Checking one by one can easily eliminate these errors.

Author Response

Response to Reviewer 4 Comments

Point 1: On line 166 to 168 and line 174 to 177, the serial number of equations are not alignment. And the size of equation which is on line 174 is not same as other equations. In equation (10), the comma after the first equation is lost.

Response 1: We apologize for the issues in the previous manuscript. We have fixed these issues in the revised manuscript.

Point 2: There is a lack of explanation of Cj*from equation (1) on line 161 to 165

Response 2: We understand the concern raised by the reviewer. The term refers to the predicted class by ensembled classifiers, and we have highlighted the same on the line after the equation, as suggested by the reviewer.

Point 3: In equation (2), it's much clearer to put π in front of the division equation.

Response 3: We have incorporated the suggested change in the revised manuscript.

Point 4:  There should be more explanation of the model architecture in the title of Figure 1 to make it easier for readers to understand. The title of Figure 2 has the same problem.

Response 4: Based on the suggestion of the reviewer, we have added detailed explanations regarding Figures 1 & 2.

Point 5 On line 230,  can’t be found in equation (9). It should probably be checked for a case error.

Response 5: We apologize for the confusion in the previous manuscript. We have rectified the issue regarding equation (9).

Point 6:  On line 315, there is an error in the parentheses of the  equation.

Response 6: Upon rechecking, the parentheses of equation  are found to be correct according to the theory.

Point 7:  The data of the tables in the article is too messy to highlight the key points.

Response 7: In response to this comment, the tables have been extended and reformatted for a clearer understanding.

Point 8:  In addition, the manuscript contains a number of typographical errors. For instance, The layout of equation (9) and equation (10), equation (11) and equation (13) don't line up. Checking one by one can easily eliminate these errors.

Response 8: We apologize for the typographical errors. We have rechecked the manuscript and ensured that there are no such issues in the revised manuscript.

Round 2

Reviewer 1 Report

Thank you for your answers, but they are not convincing.

The aim of developing new methods of data analysis is to obtain better results in relation to the solutions described in the literature. Minor improvements to the voting scheme in the classifier ensemble did not result in a significant improvement in the accuracy of Covid image classification. Similarly, in the case of XAI, the proposed solution did not result in a better understanding of the classification rules developed by the network. In particular, it has not been shown that the image areas indicated as the most important for classification contain diagnostically significant information. The lack of participation of radiologists in assessing the effectiveness of the CNN explainabilty is a major limitation of this study.

Finally, the use of historical data does not automatically lead to the conclusion that the developed methods will be effective in distinguishing pulmonary lesions caused by current coronavirus variants. The paper again lacked a discussion with radiologists about the nature of these changes and the differences in lung damage caused by historical and current virus' variants.

In conclusion, from the point of view of clinical applications, the contribution of this work is very limited. However, seeing in this work some new solutions regarding the classification process using deep networks, I recommend sending the paper to another journal devoted to machine learning or its application in image analysis.

Author Response

We thank the reviewer for the detailed and insightful review of our manuscript. Please see the attachment.
